# Ultralight, Ultraflexible, Anisotropic, Highly Thermally Conductive Graphene Aerogel Films

**DOI:** 10.3390/molecules26226867

**Published:** 2021-11-14

**Authors:** Zheng Liu, Qinsheng Wang, Linlin Hou, Yingjun Liu, Zheng Li

**Affiliations:** 1Special Equipment Safety Supervision and Inspection Institute of Jiangsu Province, National Graphene Products Quality Inspection and Testing Center (Jiangsu), 330 Yanxin Road, Huishan, Wuxi 214174, China; wqs@wxxtjy.com; 2MOE Key Laboratory of Macromolecular Synthesis and Functionalization, Department of Polymer Science and Engineering, Key Laboratory of Adsorption and Separation Materials & Technologies of Zhejiang Province, Zhejiang University, 38 Zheda Road, Hangzhou 310027, China; linlin.hou@materials.ox.ac.uk (L.H.); yilunliu@mail.xjtu.edu.cn (Y.L.); lizheng_zju@zju.edu.cn (Z.L.)

**Keywords:** graphene aerogel film, anisotropic material, thermal conductivity

## Abstract

Graphene aerogels have attracted much attention as a promising material for various applications. The unusually high intrinsic thermal conductivity of individual graphene sheets makes an obvious contrast with the thermal insulating performance of assembled 3D graphene materials. We report the preparation of anisotropy 3D graphene aerogel films (GAFs) made from tightly packed graphene films using a thermal expansion method. GAFs with different thicknesses and an ultimate low density of 4.19 mg cm^−3^ were obtained. GAFs show high anisotropy on average cross-plane thermal conductivity (K_⊥_) and average in-plane thermal conductivity (K_||_). Additionally, uniaxially compressed GAFs performed a large elongation of 11.76% due to the Z-shape folding of graphene layers. Our results reveal the ultralight, ultraflexible, highly thermally conductive, anisotropy GAFs, as well as the fundamental evolution of macroscopic assembled graphene materials at elevated temperature.

## 1. Introduction

Graphene, a monolayer of sp^2^ bonded two-dimensional (2D) carbon atoms, has gained attention over recent years [1]. The 3D graphene architectures of aerogels, foams, and sponges hold the advantages of low density, high porosity, high specific surface area, and stable mechanical properties [2,3,4,5]. Compared with conventional aerogels, anisotropic GAs have a higher degree of controllability in microstructure with direction-dependent functions. The direction-dependent physical properties make anisotropic GAs ideal for thermal shock resistance, electromagnetic interference shielding, magnetic, catalytic, and optoelectronic applications [6,7,8,9]. Graphene aerogels (GAs) thus far are generally prepared by the integration of individual graphene sheets and a template-directed approach [10]. The priority for fabricating GAs in bulk form is to avoid the stacking of graphene sheets. Integrating of individual graphene sheets from the graphene oxide (GO) dispersion needs to precisely manipulate the gelation, reduction, or cross-linking process. The balance of supramolecular interactions, including hydrogen bonding, π-π stacking, electrostatic interaction, and coordination, brings great difficulty on controlling the aerogel structure [11,12,13]. Methods based on oriented structured templates like directional-freezing [6], polymer templating [7], calcium ion-assisted unidirectional-freezing [8] have been used to make anisotropic GAs. However, the morphology of GAs produced by the template-directed approaches is determined by templates, polymers or metallic substrates, removal of which always causes collapse of the as-formed pore structure of graphene networks [14].

The thermal conductivity of single-layered graphene measured by the noncontact Raman spectroscopy technique is in the range of~3000–5300 W m^−1^ K^−1^ at room temperature [15]. It shows excellent potential as thermal management material for electronic devices [16]. To exploit the high thermal conductivity, graphene is generally in the form of macroscopic assemblies like 2D films and 3D foams. For macroscopic assembled graphene materials, their thermal conductivities are determined by sheet size, defects, assembly pattern, packing density, thermal boundary resistance, orientation, and structure continuity. When it comes to 3D assembled graphene materials, however, GAs usually are expected to be isotropic and thermal insulating: owing to the loosely packed networks of graphene sheets, which exhibit sizeable thermal boundary resistance at graphene–graphene, and graphene–air interfaces [17]. Using traditional sheet assembly and CVD growth approaches, GAs typically demonstrated low thermal conductivity from 4.7 × 10^−3^ to 1.7 W m^−1^ K^−1^ [18,19,20,21]. It is still a great challenge to fabricate GAs with oriented structures and high thermal conductivity.

The present study demonstrates ultralight, ultraflexible, anisotropic and highly thermally conductive graphene aerogel films (GAFs). Utilizing the opposite process of sheet-assembly, we employed a thermal expansion method starting from tightly packed graphene films. The resulting GAFs still kept the film appearance in the macroscopic view, forming a microscopic 3D porous network, reaching an ultralight weight of 4.19 mg cm^−3^. We studied the impact of thermal treatment on the structural change of graphene sheets and the evolution of their packing situation. We proposed a mechanism for forming 3D GAF architectures relating to the interlayer accumulation and expansion due to the decomposed gas. The GAFs show highly anisotropic conductive properties. The cross-plane thermal conductivities (K_⊥_) were only 0.3–0.7 W m^−1^ K^−1^ due to the relatively low densities. The K_||_ value showed a rapid increase from 8.98 to 53.56 W m^−1^ K^−1^ along the in-plane direction, when the density increased from 40.32 mg cm^−3^ to 150.49 mg cm^−3^. Besides, compressed GAFs with a thickness of 19 µm showed an ultrahigh elongation of 11.76% during the tensile test. This high-temperature thermal expansion is convenient access to the unique oriented 3D architectures of GAFs and attracts anisotropic thermal performances.

## 2. Materials and Methods

### 2.1. Materials

Graphene oxide (GO) was purchased from Hangzhou Gaoxi Technology Co. Ltd. (Hangzhou, China). Other chemical reagents were all analytical grade and used as received. GO dispersion was prepared using deionized water (18.2 MΏ cm).

### 2.2. Preparation of GAFs

GO films were prepared by vacuum filtration of graphene dispersion. The thickness could be controlled easily by adjusting the amount of GO dispersion. The GO film was fixed on a Teflon plate, soaked in 40% of the HI solution, and reduced to 12 h at 80 °C. Then the RGO film was rinsed with water, saturated sodium bicarbonate solution, water and ethanol solution, and dried in a vacuum of 100 °C for 12 h to obtain the reduced graphene film.

Heat treatment was carried out using a high-temperature graphitized furnace, and the RGO is placed in a graphite box, put into the medium frequency high-temperature graphite chemical furnace under the protection of argon heating treatment, heating up to the target temperature and kept for a time (500 °C, 1000 °C, 1500 °C and 2000 °C for 2 h; 3000 °C for 0.5 h). GAFs were compressed using vertical oil pressure jacks that hold for 1 h at 50 MPa.

### 2.3. Characterization

SEM images were obtained using a Hitachi S4800 field emission SEM system. XRD patterns were collected on a Philips X’Pert PRO diffractometer equipped with Cu Kα radiation (40 kV, 40 mA) with an X-ray wavelength (λ) of 1.5418 Å. Raman spectra were taken on a Renishaw in Via-Reflex Raman microscopy at an excitation wavelength of 532 nm with less than 1 mW laser power and 10 s integration time. The tensile tests were carried out on an HS-3002C mechanical testing system. GO, and graphene films were cut to 3 mm wide films. Both ends of the individual film were fixed onto clamps made by two PDMS slices with gauge lengths of 10 mm, respectively. A loading rate of 2 mm min^−1^ was applied in all mechanical tests. SEM images of the fracture section determined the cross-section area. Thermal diffusivity was measured by a Netzsch LFA 457Gerätebau GmbH, Wittelsbacherstr.laser flash thermal diffusivity apparatus. The thermal conductivity of the GAFs was determined according to Ref. [22].

## 3. Results and Discussion

The thickness of the annealed graphene film increased with the processing temperature. The cross-sections of graphene films annealed at various temperatures are presented in Figure 1. At 500 °C, the film showed a loose and layered structure with obvious interspace different from original graphene films. This interlayer spacing continuously expanded to a bubble structure at 1000 °C (Figure 1B). The system of bubbles is enclosed gas chambers composed of stacked multilayer graphene. The cross section of the graphene film annealed at 1500 °C showed a more significant porosity than the film annealed at 1000 °C. When the annealing temperature reached 2000 °C, the chamber wall became bumpy and exhibited large bubbles within the structures. At 2500 °C, stacked multilayer graphene cracked into plenty of smaller-sized bubbles. The final loose and porous structure formed accompanied more bubble walls bursting at 3000 °C (Figure 1F).

The annealing process on graphene films was followed by Raman spectroscopy (Figure 1G, detailed data are listed in Appendix A). The ID/IG value increased when the original GO films were reduced by HI acid and annealed below 1000 °C, implying the increased amount of “unorganized” carbon in the samples [23]. Then the ID/IG value decreased to 0.16 at the annealing temperature of 1500 °C. The D band was barely observed at 2000 °C or higher due to the regain of conjugated areas. The 2D band centered at 2694 and 2692 cm^−1^ for GO and HI-reduced GO, respectively, just as it did for the monolayer graphene but with a larger linewidth [24]. The wavenumbers of 2D band upshifted to 2741 cm^−1^ with the increasing annealing temperature, indicating the graphic structure’s effective recovery. The 3000 °C treatment extensively repaired the conjugated system of graphene and resulted in defect-free graphene [25]. The evolution of Raman spectra with increasing treating temperature represents the formation of a turbostratic stacking order in the direction perpendicular to the layer plane for the final GAFs.

X-ray diffraction (XRD) patterns show that the 2θ peak values of graphene films increase with the annealing temperature. Meanwhile, the corresponding d-spacing gradually approaches the value of graphite. Detailed information for GAFs annealed from 500 °C to 3000 °C are obtained in Figure 1H and Appendix A. The XRD patterns show 2θ peaks ranging from 25.35° to 26.52°, corresponding to d-spacing from 3.51 Å to 3.35 Å and crystalline domain size from 5.29 nm to 27.94 nm. d-spacing of the final GAFs (3.35 Å) is close to pristine natural graphite (3.35 Å). Raman and XRD results demonstrate that graphene films gradually regained their conjugated areas, accompany with decrease of d-spacing during the high-temperature thermal annealing.

Compared with the glistening flexible dense HI-reduced graphene film in Appendix A left, graphene films annealed at 3000 °C become loose, porous and metal-gray with no glistening. The graphene film expanded about 25-fold in the thickness direction from 40 µm to 1 mm, while the film’s 15.1% diameter decrease was also observed. The volume of the GAF was increased by 1800% (measured density of 23 mg cm^−3^), while it still kept the film appearance in macroscopic view. Besides, resulted GAFs showed an ultimate ultralight weight of 4.19 mg cm^−3^. Vertical expansion results in diameter shrinkage, because the conformation of sequential horizontal stacked graphene layers transformed into deformation along the in-plane direction. Observed from the SEM images of a film cross-section was made by tear opening, the GAFs burst into a loose and porous structure (Appendix A). Graphene sheets in the same layer still kept stacking on each other and formed a wave-shaped deformation along the in-plane direction.

This expansion could be mainly ascribed to the generation and development of H_2_O and CO_2_ evolved during thermal annealing. By monitoring the weight loss of the graphitization, apparent weight loss of HI-reduced graphene films reaching 30 wt.% was observed. The elemental analysis data indicate that the HI-reduced graphene films still kept some of the oxygen-containing groups (Appendix A). These groups would decompose during thermal annealing. Besides, reduced graphene has a molecular shielding effect, which results in the isolation and accumulation of decomposed gases. With increasing temperature, the concentrated gases reached a high pressure of solid power to expand against the packed graphene layers until gases leak through the films (Figure 2). At higher temperatures over 2000 °C, graphene sheets could gain more energy to overcome the interlayer van der Waals forces [26]; thus, relaxation, slippage, stretching, and buckling of graphene sheets occurred [27]. At 3000 °C, the gas constraint in the stacked chamber walls reached the pressure to break interlayer packing, giving smaller pores (Figure 1F).

As the expansion mechanism discussed in this paper, the reduction of film thickness should lower the possibility of forming stacked graphene that blocks gas emissions and encloses gas chambers. Therefore, films with different thicknesses were also treated by the same procedure to study the expansion process. As shown in Figure 3, HI-reduced graphene films with thicknesses of 2 µm, 5 µm and 40 µm annealed at 3000 °C were expanded to 3 µm (150%), 20 µm (400%) and 1 mm (2500%). Even the graphene film of 2 µm gained a 50% thickness increase, and the porous structure is more like a gas chamber in 40 µm annealed graphene film. Aggregation and expansion of released gas during the high-temperature annealing caused the loose and porous structure of GAFs. Compared with graphene aerogels that prepare by bottom-up strategies, the thermal expansion method in this work has many advantages. Firstly, graphene in GAFs would regain the conjugated structure ensuring its further applications; secondly, graphene sheets in GAFs showed a much higher orientation due to the layered structure of original films; in addition, GAFs offer a very regular microstructure.

The high annealing temperature of 3000 °C makes graphene sheets regain their conjugate structure (domains), which are highly flexible and easy to stack. GAFs with different thicknesses were compressed into compact films using the method for making graphite devices. The compression was carried out using a plate vulcanizing machine under 100 MPa pressure. Compressed GAFs showed high flexibility (Figure 4). Even when the film was folded in half, the surface of compressed GAF can be recovered by smoothing. Tensile tests revealed ultimate strains of 11.76%, 6.05%, and 5.17% for compressed GAFs with thicknesses of 19 µm, 5 µm, and 3 µm, respectively (Figure 4F). The stress-strain curve of 3 µm compressed GAFs exhibited apparent steplike periodic features, while the 19 µm film also produced a steplike curve on a much smaller scale (Appendix A). Cross-sectional images of compressed GAFs illustrate the multilayer graphene of gas chambers piled up with large-scale bending and Z shape folding under compression (Appendix A). The ultrahigh elongation and flexibility are mainly originated from the stretching and unfolding of the deformed graphene sheets.

Compressed GAFs are very different from traditional graphite films made by compression molding. Firstly, the compressed graphene films are smooth and uniform with a silvery metallic luster, while graphite films are black. Secondly, the graphene film is very flexible, while the graphite film is fragile with insufficient flexibility. Thirdly, in the microcosmic view, compared with the unoriented graphite crystalline domain in graphite films, graphene layers in GAFs are highly oriented in the vertical direction, which is more evident in thicker ones (Appendix A). The layered structure of compressed GAFs is indistinct, other than the instinct layered structure of graphene oxide or HI-reduced graphene films. Large graphene sheets regained a conjugated structure and displayed higher flexibility and toughness, which resulted in the slippage and haul out. The flexibility and compressibility of GAFs make it an ideal shape-compatible material for thermal interfacial conducting.

Single-layered graphene has an extremely high thermal conductivity of 5300 W m^−1^ K^−1^. Practically, macroscopic graphene aerogels are usually thermally insulating because of the high porosity and low air cross-convection. Statistically, graphene sheets in normal 3D monoliths are oriented randomly, which results in isotropic thermal diffusivity in macroscopic graphene aerogels. In this study, high-temperature thermal annealing and expanding results in an anisotropic structure and thermal transport properties. The final GAFs formed a structure in which graphene layers are continuously paralleled in the vertical direction and enclose many air chambers. The incorporation of lateral orientation and high structural integrity contributes to the enhancement of phonon transmission in the lateral plane for thermal transport.

The anisotropy of thermal conductivity was characterized using the laser flash method. Among non-steady-state methods, laser-flash thermal diffusivity measurement is widely used for its low limitation on sample size, rapid and stable measurement, simple calculation process, adjustable laser pulse energy and width, strong applicability, and wide measurement temperature range. The temperature-dependent in-plane and cross-plane thermal diffusivities (α_||_ and α_⊥_) for a GAF with a thickness of 1 mm and a compressed graphite plate from 25 °C to 500 °C are shown in Figure 5A,B. Over the whole temperature range, the thermal conductivity in both directions decreased monotonically. The measured values were 75.1 mm^2^ s^−1^ (α_||_) and 421.5 mm^2^ s^−1^ (α_||_) for GAF at 25 °C, while 23.5 mm^2^ s^−1^ (α_⊥_) and 106.3 mm^2^ s^−1^ (α_||_) for GAF at 25 °C. The anisotropy (α_⊥_/α_||_) decreased monotonically from 5.61 at 25 °C to 4.52 at 500 °C. In contrast, the compressed graphite plate showed slightly higher α_||_ than α_⊥_, and, thus, very low anisotropy α_⊥_/α_||_ from 1.29 to 1.07. This result indicates that the GAFs possess a highly aligned structure and exhibit much higher anisotropy than the isotropic compressed graphite plate.

The density-dependent thermal performance for GAFs was further studied using a stepwise uniaxial compression. The cross-plane thermal diffusivities (α_⊥_) fluctuated at 60–120 mm^2^ s^−1^ without evident tendency till the thickness of GAFs decreased below the detective limit (Figure 5C). Meanwhile, the calculated cross-plane thermal conductivities (K_⊥_) were only 0.3–0.7 W m^−1^K^−1^ due to the low densities of 4~10 mg cm^−3^. On the other hand, the α_||_ value rapidly increased from 293.46 to 448.23 mm^2^s^−1^, and the corresponding thermal conductivities (K_||_) were from 8.98 to 53.56 W m^−1^K^−1^ with the density increasing from 40.32 to 150.49 mg cm^-3^ (Figure 5D). The in-plane thermal conductivities linearly increased with increasing densities. The anisotropy coefficient of thermal conductivities (K_||_/K_⊥_) was 29.93 for GAF of 40.32 mg cm^−3^ and 76.51 for GAF of 150.49 mg cm^-3^. Final compressed GAFs had a thermal conductivity of 734 W m^−1^ K^−1^ at a density of 1.59 g cm^−3^. Besides, the electrical conductivity of the compressed GAFs was ~10^5^ S m^−1^, much higher than that of HI-reduced graphene films [1].

## 4. Conclusions

In summary, 3D graphene aerogel films (GAFs) prepared from tightly packed graphene films were successfully developed by a facial high-temperature thermal expansion method. The isolation, accumulation, and expansion of decomposed gases combined with relaxation, slippage, and buckling of graphene sheets at 3000 °C gave a loose and porous structure to the GAFs. The 3D graphene film architecture exhibits an ultralight weight of 4.19 mg cm^−3^, anisotropy coefficient of thermal conductivity (K_||_/K_⊥_) ranging from 29.93 to 76.51, and an ultrahigh elongation of 11.76% for 19 µm-thick graphene films after uni-axial compression. The ultralight, ultraflexible and highly thermally conductive anisotropy of GAFs should lend them to many applications in thermal interfacial conducting, electromagnetic shielding and energy storage.

## Figures and Tables

**Figure 1 molecules-26-06867-f001:**
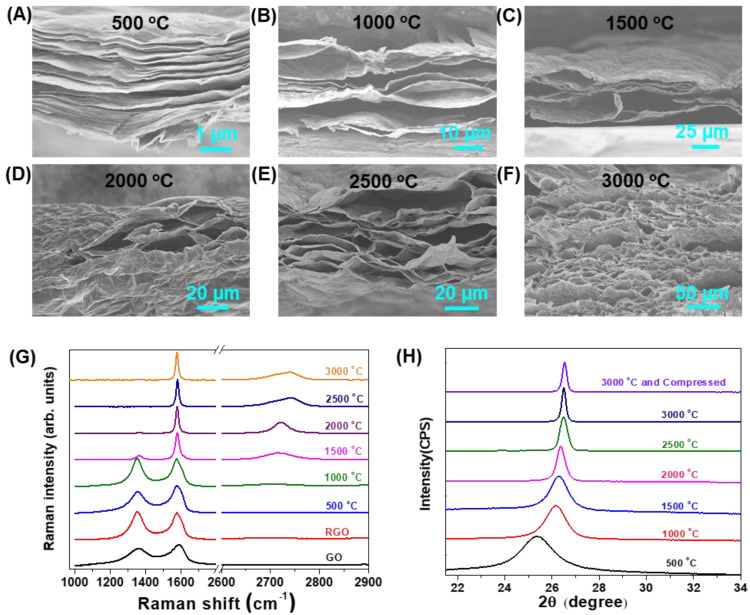
(**A**–**F**) SEM images of the cross-sections of graphene films after different annealing temperatures. (**G**) Raman spectra and (**H**) XRD patterns of GO films, HI-reduced GO films, and graphene films treated under different annealing temperatures.

**Figure 2 molecules-26-06867-f002:**
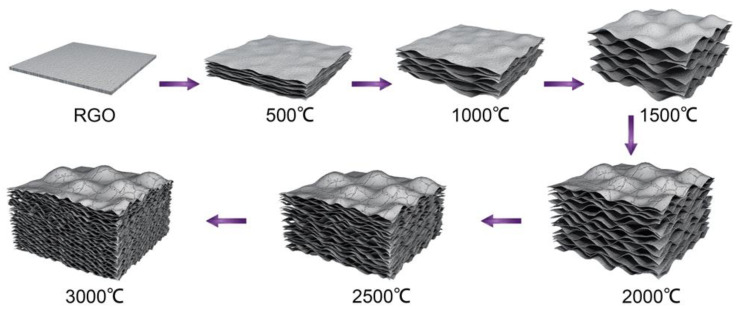
Formation of graphene aerogel films (GAFs).

**Figure 3 molecules-26-06867-f003:**
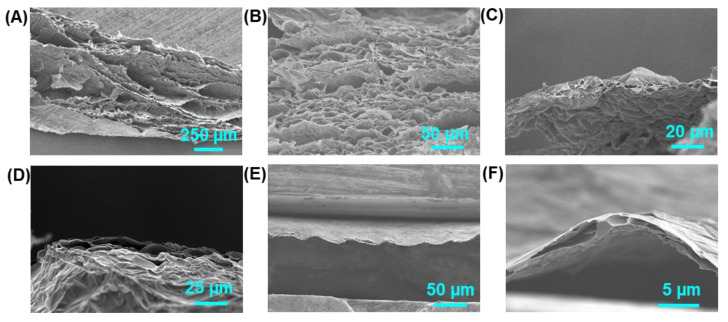
SEM images of cross-sections of GAFs with thicknesses of (**A**,**B**) 1 mm, (**C**,**D**) 20 µm, and (**E**,**F**) 3 µm.

**Figure 4 molecules-26-06867-f004:**
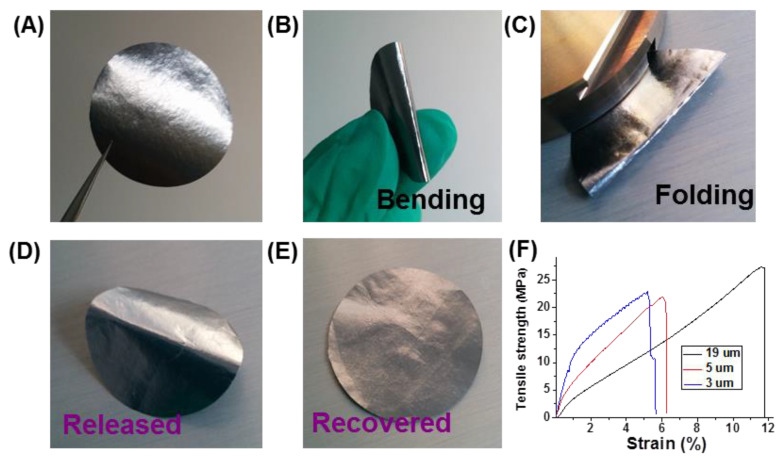
(**A**,**B**) A compressed GAF with glistening and high flexibility. (**C**) The compressed GAF is folded in half. (**D**) Release of folded compressed GAF, and (**E**) recovery by smoothing. (**F**) Typical mechanical measurements under tensile loading for compressed GAFs with different thicknesses.

**Figure 5 molecules-26-06867-f005:**
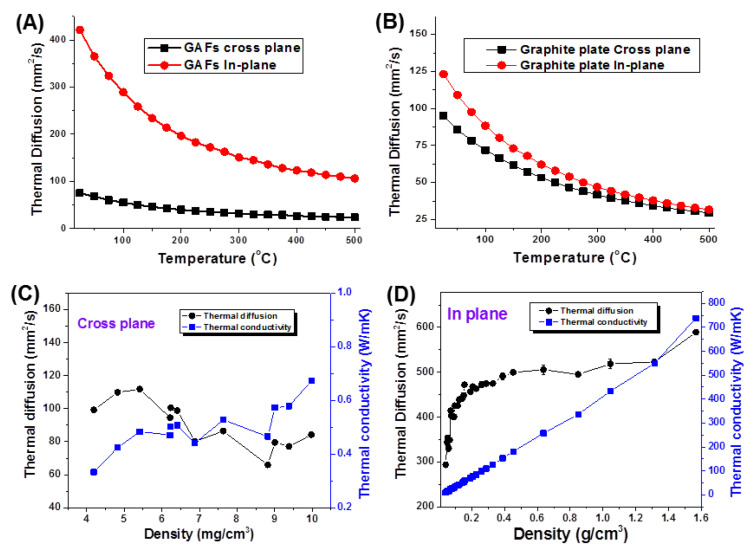
In-plane and cross-plane thermal diffusivity vs. temperature for typical (**A**) GAFs and (**B**) compressed graphite plates. The cross-plane (**C**) and in-plane (**D**) thermal diffusivity, thermal conductivities vs. densities for GAFs.

## Data Availability

The data presented in this study are available in Appendix A.

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
