# Peer review of "Ultralight, Ultraflexible, Anisotropic, Highly Thermally Conductive Graphene Aerogel Films"

_molecules, 2021, doi:10.3390/molecules26226867_

Round 1

Reviewer 1 Report

This work reports the production of reduced graphene oxide (rGO) 3D porous films by thermal annealing in Argon atmosphere. The influence of the annealing temperature (in the range 500°C – 3000°C) on the mechanical properties and thermal properties is investigated and correlated with the morphology and structure. Although the reported results are of some interest from the point of view of applied research, major revisions are suggested before publication. Please note that I was not able to find the Supporting Information file anywhere, so I apologize in advance if my comments do not take the supporting material into account.

Specifically, my comments are:

  • The authors suggest a simple thermal treatment for inducing strong thermal conductivity anisotropy in the 3D rGO films. The authors should expand the discussion of these results with a more complete comparison with the existing literature
  • If possible, it would be interesting to provide some insight into the chemical composition of the rGO film annealed at different temperatures, for example with X-ray Photoelectron Spectroscopy.
  • The authors should provide some insight into the laser flesh method for measuring thermal diffusivity, with its advantages and limitations compared to other methods. Moreover, they should explicitly state how they obtain thermal conductivity from thermal diffusivity measurement.
  • There are some English errors that should be fixed
  • If possible, the introduction should be slightly expanded in the discussion of thermally anisotropic graphene-based porous materials.

Author Response

Dear reviewer

Thank you very much for your positive and constructive comments on our manuscript. Accordingly, we have well revised it, and we believe that this version has been highly improved.

The point-to-point responses are listed as follows:

Comments 1 The authors suggest a simple thermal treatment for inducing strong thermal conductivity anisotropy in the 3D rGO films. The authors should expand the discussion of these results with a more complete comparison with the existing literature

Response: Thanks for this comment. Comparison between our graphene aerogel films and other graphene aerogel have been added in the revised manuscript. (Page 5, line 187)

Comments 2 If possible, it would be interesting to provide some insight into the chemical composition of the rGO film annealed at different temperatures, for example with X-ray Photoelectron Spectroscopy.

Response: Thanks for this comment. We were also very concerned about changes in graphene chemical composition, XPS of graphene films after different anneal temperatures have been carried out. But the results showed that after 1500 degrees, the XPS peak of the sample was left with only carbon and oxygen elements, and no further changes were observed, many articles have reported this results. So we did not put this result into this manuscript.

Comments 3 The authors should provide some insight into the laser flesh method for measuring thermal diffusivity, with its advantages and limitations compared to other methods. Moreover, they should explicitly state how they obtain thermal conductivity from thermal diffusivity measurement.

Response: Thanks for this comment. The description of the characteristics of laser flesh method has been added to the revised manuscript (Page 7, line 241). Thermal conductivity of the GAFs were determined according to Ref. (Adv. Funct. Mater. 2014, 24, 4222–4228.) which have been put into the revised manuscript (2.3 Characterization, Page 3, line 105).

Comments 4 There are some English errors that should be fixed

Response: Thanks for this comment. We have revised our manuscript according to this comment.

Comments 5 If possible, the introduction should be slightly expanded in the discussion of thermally anisotropic graphene-based porous materials.

Response: Thanks for this comment. Discussion on thermally anisotropic graphene-based porous materials have been put into the revised manuscript (1 Introduction, Page 1, line 30 and line 41).

Reviewer 2 Report

In this manuscript, the authors developed ultralight, anisotropic highly thermally conductive graphene aerogel films by a facial high temperature thermal expansion method. The impact of thermal treatment on structural change of graphene sheets and the evolution of their packing situation was studied. This work seems interesting. I’d like to recommend the manuscript for publication once the following comments are addressed:

  1. Due to different processing methods, the compressed GAFs and GAFs possess different properties. Figure 4 shows the flexibility and tensile properties of compressed GAFs, but there is no relevant characterization about GAFs. According to the theme of this work, the article should focus on GAFs instead of the compressed GAFs, and the compressed GAFs exhibit high flexibility does not mean that GAFs possess high flexibility, so could you add more relevant analysis and discussion on GAFs?
  2. I noticed that the compressed graphene films are silvery metallic luster while graphite films are black, this is an interesting experimental phenomenon, could you analyze the mechanism?
  3. You mentioned that the electrical conductivity of the compressed GAFs is ~105 S m-1, which is much higher than that of HI reduced graphene films, could you discuss this in more detail, and why the compressed GAFs possess higher electrical conductivity?
  4. Could you compare the thermal conductivity of HI reduced graphene films and GAFs, which cloud more directly illustrate the effect of the structure of GAFs on the thermal conductivity.
  5. In the last paragraph on page 3, “In comparison with… in figure 2A left…”. I did not see the corresponding content in Figure 2, please check carefully whether the relevant Figures were missing.
  6. There are many writing errors in the manuscript, for example, on page 4, line 135, it seems to be “Figure 2”, instead of “Scheme 1”; In Figure 5D, the unit of abscissa should be “mg/cm3”, instead of “g/cm3”; On page 6, line 206, it seems to be “α⊥”, instead of “α=”… Please check the manuscript carefully to make sure there are no writing errors.
  7. The authors could add the following reference which would again increase the interest to general functional gel readers: Chemical Society Reviews‚ 2021, 50, 8319-8343; Advanced Functional Materials‚ 2021‚ 31, 2103391;ACS Applied Materials & Interfaces‚ 2021‚13,7617-7624.

Author Response

Dear reviewer

Thank you very much for your positive and constructive comments on our manuscript. Accordingly, we have well revised it, and we believe that this version has been highly improved.

The point-to-point responses are listed as follows:

Comments 1  Due to different processing methods, the compressed GAFs and GAFs possess different properties. Figure 4 shows the flexibility and tensile properties of compressed GAFs, but there is no relevant characterization about GAFs. According to the theme of this work, the article should focus on GAFs instead of the compressed GAFs, and the compressed GAFs exhibit high flexibility does not mean that GAFs possess high flexibility, so could you add more relevant analysis and discussion on GAFs?

Response: Thanks for this comment. GAFs can not be fixed on the fixture for mechanical testing stably due to its loose structure. We have tried many times and have not obtained valid data. We found that the thermally annealed GAFs are very easy to be compressed due to the regain of the conjugated structure. Besides, the test of the film after compression can well reflect the changes and advantages of GAFs in terms of structure, so we discussed the compressed GAFs in the article.

Comments 2  I noticed that the compressed graphene films are silvery metallic luster while graphite films are black, this is an interesting experimental phenomenon, could you analyze the mechanism?

Response: Thanks for this comment. There are a large number of free electrons similar to metals in the structure of graphite and graphene, which can reflect electromagnetic waves in the visible light band to form the so-called metallic luster. Graphene in compressed GAFs are highly oriented, which will form a large reflective surface on the surface of the film result in silvery metallic luster. Heat dissipation films prepared by high-temperature graphitization using PI as raw materials have similar properties. Graphite grains are often arranged in a random orientation in the graphite film, and it is difficult to form a complete reflective surface, so it appears black.

Comments 3  You mentioned that the electrical conductivity of the compressed GAFs is ~105 S m-1, which is much higher than that of HI reduced graphene films, could you discuss this in more detail, and why the compressed GAFs possess higher electrical conductivity?

Response: Thanks for this comment. Although the graphene film reduced by HI chemically repairs some defects of graphene oxide, the recovery of its conjugated structure is not perfect. GAFs After the high-temperature graphitization of the GAFs, all heteroatoms other than carbon have been removed, and the conjugated structure of graphene has also been greatly restored. The compression process greatly increases the contact between the graphene sheets and reduces the contact resistance between graphene sheets, so it showed a good conductivity.

Comments 4  Could you compare the thermal conductivity of HI reduced graphene films and GAFs, which cloud more directly illustrate the effect of the structure of GAFs on the thermal conductivity.

Response: Thanks for this comment. We have tried to tested the HI-reduced film during the research process, but the test results were not stable. After the test, we found that the part of the film directly flashed by the laser turned black, which may be related to the instability of the HI-reduced film.

Comments 5  In the last paragraph on page 3, “In comparison with… in figure 2A left…”. I did not see the corresponding content in Figure 2, please check carefully whether the relevant Figures were missing.

Response: Thanks for this comment. The ‘figure 2A left’ is indeed a mistake. The corresponding picture is in Supplementary Information Figure S1A. We have corrected it in the revised manuscript.

Comments 6  There are many writing errors in the manuscript, for example, on page 4, line 135, it seems to be “Figure 2”, instead of “Scheme 1”; In Figure 5D, the unit of abscissa should be “mg/cm3”, instead of “g/cm3”; On page 6, line 206, it seems to be “α⊥”, instead of “α=”… Please check the manuscript carefully to make sure there are no writing errors.

Response: Thanks for this comment. In Figure 5D, the unit of abscissa is “g/cm3”, because as the compression ratio increases, the density of the final film reaches the range of “g/cm3”. Other errors have been corrected in the revised manuscript.

Comments 7  The authors could add the following reference which would again increase the interest to general functional gel readers: Chemical Society Reviews‚ 2021, 50, 8319-8343; Advanced Functional Materials‚ 2021‚ 31, 2103391;ACS Applied Materials & Interfaces‚ 2021‚13,7617-7624.

Response: Thanks for this comment. We have put related references into the revised manuscript.

Reviewer 3 Report

In this manuscript, graphene aerogel films were prepared by a thermal expansion method. Authors analysed the films.

The experiments and discussion can add value to the literature. 
The output of the study can be useful in applications like energy storage or electromagnetic shielding.

Below are my comments/suggestions:
In line 2; shouldn't there be a comma or 'and' after the word "anisotropic"?

In Section-4.3: What was the laser WL, power, integration time of Raman Spectroscopy?

Section-4 can be moved to the Line 72 as the Section-2 which can be related to the "Experiments".

Which concentration range did you use to prepare GO films by GO:deionized water for the vacuum filtration? 

It may be important to provide information on how expensive GOF may be to manufacture.

In Line 60: "...3D mirco-porous network .." or "...m-3."

In Line 115: Where is figure 2A left?

In Line 142-143: Figure 3(F) was not mentioned?

Author Response

Dear reviewer

Thank you very much for your positive and constructive comments on our manuscript. Accordingly, we have well revised it, and we believe that this version has been highly improved.

The point-to-point responses are listed as follows:

Comments 1 In line 2; shouldn't there be a comma or 'and' after the word "anisotropic"?

Response: Thanks for this comment. We have revised it in the revised manuscript according to this comment.

Comments 2 In Section-4.3: What was the laser WL, power, integration time of Raman Spectroscopy?

Response: Thanks for this comment. We have corrected it in the revised manuscript according to this comment.

Comments 3 Section-4 can be moved to the Line 72 as the Section-2 which can be related to the "Experiments".

Response: Thanks for this comment. We have moved this section in the revised manuscript (Page 2, line 76).

Comments 4 Which concentration range did you use to prepare GO films by GO:deionized water for the vacuum filtration? 

Response: Thanks for this comment. The concentration of GO disspersed in deionized water is around 5 mg/mL.

Comments 5 It may be important to provide information on how expensive GOF may be to manufacture.

Response: Thanks for this comment. The manufacturing cost of GOF is determined according to the cost of GO. With the large-scale preparation and process improvement of GO, the cost of preparing GOF has been reduced by 70%-80% compared with 5 years ago.

Comments 6 In Line 60: "...3D mirco-porous network .." or "...m-3."

Response: Thanks for this comment. We have used “microscopic 3D porous network” in the revised manuscript according to this comment.

Comments 7 In Line 115: Where is figure 2A left?

Response: Thanks for this comment. The ‘figure 2A left’ is indeed a mistake. The corresponding picture is in Supplementary Information Figure S1A. We have corrected it in the revised manuscript.

Comments 8 In Line 142-143: Figure 3(F) was not mentioned?

Response: Thanks for this comment. We have corrected it in the revised manuscript according to this comment.

Reviewer 4 Report

1. please provide reported values with their uncertainties, or at least please estimate them and unify the notation. For example, in the abstract, the Authors wrote the following sentence: "The GAFs were anisotropic since the cross-plane thermal conductivity (K⊥) was 0.3-0.7 W m-1K-1, while the in-plane thermal conductivity (K||) was 8.98-53.56 Wm-1K-1 for density of 40.32-150.49 mg cm-3." Please read the "Evaluation of measurement data — Guide to the expression of uncertainty in measurement" (https://www.bipm.org/documents/20126/2071204/JCGM_100_2008_E.pdf/cb0ef43f-baa5-11cf-3f85-4dcd86f77bd6)

2. since the laser flash method typically assumes homogenous material and the Authors report on the anisotropy of the thermal conductivity, it is required to describe the principles of the Authors' method and models. Lack of methodology is a major drawback of this work.

3. please consider adding a Table in which reported data can be compared to other literature data. It is always a good strategy to collate own results to the literature.  Such information immediately shows the novelty and significance of the work.

Other minor remarks:

4. Please consider coordinating data presented in Figures 1g and 1h. It will be nice to have them corresponding / in the same order.

5. There is a lack of arrow in Figure 2 between 1500 oC and 2000 oC

6. Please check the caption of Figure 3.

7. The symbols related to the in-plane and cross-plane thermal conductivities are written once as subscript (like in the abstract) and once as normal letters (e.g., lines 65-66).

Author Response

Dear reviewer

Thank you very much for your positive and constructive comments on our manuscript. Accordingly, we have well revised it, and we believe that this version has been highly improved.

The point-to-point responses are listed as follows:

Comments 1. please provide reported values with their uncertainties, or at least please estimate them and unify the notation. For example, in the abstract, the Authors wrote the following sentence: "The GAFs were anisotropic since the cross-plane thermal conductivity (K⊥) was 0.3-0.7 W m-1K-1, while the in-plane thermal conductivity (K||) was 8.98-53.56 Wm-1K-1 for density of 40.32-150.49 mg cm-3." Please read the "Evaluation of measurement data — Guide to the expression of uncertainty in measurement" (https://www.bipm.org/documents/20126/2071204/JCGM_100_2008_E.pdf/cb0ef43f-baa5-11cf-3f85-4dcd86f77bd6)

Response: Thanks for this comment. The thermal diffusivity test process by the laser flash method has very good stability and repeatability, and the thermal conductivity calculated based on this is also the same. We use average thermal conductivity in our work, which is described in the revised manuscript according to your comment. Also, we have revised the discussion on the anisotropy of the GAFs in the revised manuscript.

Comments 2. since the laser flash method typically assumes homogenous material and the Authors report on the anisotropy of the thermal conductivity, it is required to describe the principles of the Authors' method and models. Lack of methodology is a major drawback of this work.

Response: Thanks for this comment. The description of the characteristics of laser flesh method has been added to the revised manuscript (Page 7, line 241). Thermal conductivity of the GAFs were determined according to Ref. (Adv. Funct. Mater. 2014, 24, 4222–4228.) which have been put into the revised manuscript (2.3 Characterization, Page 3, line 105).

Comments 3. please consider adding a Table in which reported data can be compared to other literature data. It is always a good strategy to collate own results to the literature. Such information immediately shows the novelty and significance of the work.

Response: Thanks for this comment. Because there is not much work on the anisotropy of thermal conductivity of GAs, we did not use this Table in the article.

Comments 4. Please consider coordinating data presented in Figures 1g and 1h. It will be nice to have them corresponding / in the same order.

Response: Thanks for this comment. We have revised it in the revised manuscript according to this comment.

Comments 5. There is a lack of arrow in Figure 2 between 1500 oC and 2000 oC

Response: Thanks for this comment. We have corrected it in the revised manuscript according to this comment.

Comments 6. Please check the caption of Figure 3.

Response: Thanks for this comment. Thanks for this comment. We have corrected it in the revised manuscript according to this comment.

Comments 7. The symbols related to the in-plane and cross-plane thermal conductivities are written once as subscript (like in the abstract) and once as normal letters (e.g., lines 65-66).

Response: Thanks for this comment. We have corrected it in the revised manuscript according to this comment.

Round 2

Reviewer 1 Report

The authors have fixed most of the issues which were discussed in the previuos review process, therefore the paper is suitable for publication. However, regarding the English language, some minor changes are still needed before publication.

Reviewer 4 Report

.